# The Association of Age, Sex, and BMI on Lower Limb Neuromuscular and Muscle Mechanical Function in People with Multiple Sclerosis

**DOI:** 10.3390/biomedicines12050971

**Published:** 2024-04-28

**Authors:** Anne Geßner, Maximilian Hartmann, Katrin Trentzsch, Heidi Stölzer-Hutsch, Dirk Schriefer, Tjalf Ziemssen

**Affiliations:** Center of Clinical Neuroscience, Neurological Clinic, University Hospital Carl Gustav Carus, TU Dresden, Fetscherstr. 74, 01307 Dresden, Germany; anne.gessner@uniklinikum-dresden.de (A.G.); maximilian.hartmann@uniklinikum-dresden.de (M.H.); katrin.trentzsch@uniklinikum-dresden.de (K.T.); heidi.stoelzer-hutsch@uniklinikum-dresden.de (H.S.-H.); dirk.schriefer@uniklinikum-dresden.de (D.S.)

**Keywords:** multiple sclerosis, countermovement jump, muscle mechanical function, neuromuscular function, lower limb assessment

## Abstract

(1) Background: The countermovement jump (CMJ) on a force plate could be a sensitive assessment for detecting early lower-limb muscle mechanical deficits in the early stages of multiple sclerosis (MS). CMJ performance is known to be influenced by various anthropometric, physiological, and biomechanical factors, mostly investigated in children and adult athletes. Our aim was to investigate the association of age, sex, and BMI with muscle mechanical function using CMJ to provide a comprehensive overview of lower-limb motor function in people with multiple sclerosis (pwMS). (2) Methods: A cross-sectional study was conducted with pwMS (N = 164) and healthy controls (N = 98). All participants performed three maximal CMJs on a force plate. Age, sex, and BMI were collected from all participants. (3) Results: Significant age, sex, and BMI effects were found for all performance parameters, flight time, and negative and positive power for pwMS and HC, but no significant interaction effects with the group (pwMS, HC) were detected. The highest significant effects were found for sex on flight time (η^2^ = 0.23), jump height (η^2^ = 0.23), and positive power (η^2^ = 0.13). PwMS showed significantly lower CMJ performance compared to HC in middle-aged (31–49 years), with normal weight to overweight and in both women and men. (4) Conclusions: This study showed that age, sex, and BMI are associated with muscle mechanical function in pwMS and HC. These results may be useful in developing reference values for CMJ. This is a crucial step in integrating CMJ into the diagnostic assessment of people with early MS and developing individualized and effective neurorehabilitative therapy.

## 1. Introduction

Multiple sclerosis (MS) is a chronic inflammatory disease characterized by heterogeneity of symptoms and pathological mechanisms [1]. Deficits in neuromuscular function (i.e., the interaction between the nervous and the muscular system) and decrements in mechanical function of the lower-limb muscles (i.e., muscle strength, muscle power, and explosive muscle strength) are key symptoms of the disease as they are associated with impaired activities of daily living and quality of life [2,3]. Neurophysiologically, people with MS (pwMS) show lower voluntary muscle activation in terms of neuromuscular function, as well as increased fatigue attributable to the known MS pathophysiology in the central nervous system [4]. Studies indicate that reduced lower-limb muscle strength and power negatively influence the functional ability of the lower limb in pwMS in walking performance, stair climbing, and balance [3,5]. A decrease in lower-limb muscle strength, power, and rate of force development is frequently observed in pwMS compared to healthy controls (HC), especially during fast dynamic muscle contractions [2]. Muscle mechanical function may therefore serve as a particularly useful outcome for disability as it may be sufficiently sensitive to detect muscle impairment in the early stages of disease and in pwMS presenting with low disability [6].

However, lower-limb muscle mechanical function is not only associated with the degree of disability in MS but also with other factors such as age, sex, and body mass index (BMI). To identify, specify, and monitor sensitive subtle neuromuscular and muscle mechanical deficits, it is crucial to monitor all these confounding factors for the management of optimal disease-modifying and symptomatic treatment. The effect of age on muscle strength in MS can be complex. Various studies have shown that a decrease in muscle strength can be age-related [7,8]. Stagsted et al. show that the combined effects of MS and ageing result in a significant decrease in lower-limb muscle strength, which is associated with a decline in physical function [2]. Regarding BMI, its association with muscle strength or other symptoms in pwMS has not been studied, only the physical composition of muscle mass and body fat [9,10]. BMI increases gradually throughout most of adult life and loss of muscle mass begins between the ages of 30 and 40 and continues into old age [11]. However, lower-limb muscle power has been shown to be positively influenced by BMI more so in men than in women [12]. The proportion of women and men who develop MS and the influence of sex on the course of the disease have been studied. Both the central nervous system and the immune system have been shown to have sex-specific differences, which means that there are many variations in the symptoms that occur. Therefore, further studies investigating the association of sex and MS are needed to understand the sex differences in the incidence and severity of MS [13].

Among the many methods being used to evaluate lower-limb muscle mechanical function, e.g., manual muscle function tests and isokinetic dynamometry, the countermovement jump (CMJ), a vertical maximal jump, presents a new assessment in MS, specifically to measure the decrements of rapid dynamic contractions [2,5,14]. In our previous study, we showed that the CMJ can detect lower-limb motor deficits below the clinical threshold of the Expanded Disability Status Scale (EDSS) in the early stage of MS [14]. This functional assessment of muscle activity combines muscle strength, coordination, and balance and provides a simulation of everyday movements as it is based on the principal of the stretch-shortening cycle (SSC) that occurs during natural movements such as walking [14,15]. In addition, a positive relationship between lower-limb muscle strength and CMJ has been demonstrated in many studies [16,17,18]. CMJ performance is known to be influenced by various anthropometric, physiological, and biomechanical factors, mostly investigated in children and adult athletes [19,20,21,22]. Some studies have focused on the effect of sex on CMJ, finding that males showed significantly higher jumping performance than females [23,24,25]. Men jump approximately 24 to 27% higher than women [26]. Considering the effect of age, children show an increase and adults a decrease in CMJ performance with increasing age [19].

To date, there have been no studies that have investigated the CMJ in association with age, sex, and BMI in pwMS. Characterization of CMJ in association with age, sex, and BMI is necessary for the individual interpretation of early deficits in neuromuscular and muscle mechanical function and the integration of the CMJ into the diagnostic assessment in pwMS [27,28]; it enables individualized and effective neurorehabilitative therapy strategies for pwMS. Our primary aim in this study was to characterize CMJ performance in pwMS and HC in relation to age, sex, and BMI and to provide reference values for CMJ performance in pwMS.

## 2. Materials and Methods

### 2.1. Participants

A cross-sectional study was conducted at the MS Center at the Center of Clinical Neuroscience of the Department of Neurology, University Hospital Carl Gustav Carus, Dresden, Germany. Healthy controls (HC) without any neurological disease, who were age-matched to pwMS, were recruited. Recruitment took place from April 2021 to September 2022.

All participants provided written informed consent for the study. The study received approval from the local ethics committee (BO-EK-320062021). The inclusion criteria for pwMS were as follows: (a) confirmed MS diagnosis; (b) EDSS score between 0 and 3.0; (c) age from 18 to 65 years; (d) ability to walk without aid and rest for ≥500 m; and (e) to perform heel rise, stand on heels, and squats. Exclusion criteria for this study were orthopaedic and surgical conditions that could affect jumping ability, fear of falling while jumping, and current pregnancy.

### 2.2. Measures and Procedures

#### 2.2.1. Age, Sex, and Body Mass Index

Age, sex, and BMI were collected from all participants. BMI was calculated as weight in kg divided by height in m^2^ and classified as normal weight (18–25), overweight (25–30), or obese (>30). As most people are diagnosed with MS between the ages of 20 and 40 [29,30], the following age categories were chosen: young (18–30 years), middle-aged (31–49 years), and old (50–65 years). In terms of sex, the participants were binary classified as male or female.

#### 2.2.2. Expanded Disability Status Scale

Certified raters applied the EDSS to examine the neurological clinical status at pwMS, which is the most commonly used disability scale in MS [31,32]. As part of the examination, it evaluates seven functional systems and ambulation.

#### 2.2.3. Countermovement Jump

On a single force plate, all participants completed three maximal CMJ jumps without arm swing. For all three jumps, the mean values of the individual parameters were used for the statistical analyses. All participants completed a practice jump prior to data collection in order to reduce potential insecurities or errors during the test. Prior to the jumps, the physiotherapist provided verbal and visual instructions to each participant on the correct jumping technique.

For the CMJ, participants were instructed to jump as high as possible with their hands on their hips and legs fully extended during the flight phase of the jump (Figure 1). Participants rested in a standing position for approximately 5 s between jumps. Any CMJs that were inadvertently performed with arms swinging or legs tucked during the flight phase of the jumps were excluded. Jumping trials were performed in socks and everyday clothing.

### 2.3. Data Collection

A portable single force plate from AMTI (Advanced Mechanical Technology Inc., Watertown, MA, USA, AccuPower-O) measured the three-dimensional ground reaction forces (Fx, Fy, Fx) and force moments (Mx, My, Mz) during the CMJ at a frequency of 1000 Hz. This force plate, measuring 1016 × 762 × 127 mm (length × width × height), was positioned on a concrete floor and has an internal amplifier. Force plates are the gold standard and a valid instrument with which to record the vertical ground reaction force for vertical jumps [33,34]. AMTI’s force plates are strain-gauge-based and have a spring or deformation body. They have the benefits of high measurement accuracy and the ability to compensate for unwanted effects such as temperature dependence of the zero point, bending moment effects, and shear force effects [35]. The participant’s body weight was measured using the force plate in the weight phase prior to the start of the braking phase of the CMJ. A threshold value of >5 N from the measured body weight before the jump was used to define the start of the CMJ movement. The end of the countermovement phase was defined as the point when the COM position reached its lowest depth. Temporal, kinetic, and performance jump parameters during the CMJ performance were recorded with a dedicated software for biomechanical analysis (AccuPower Solutions, Version 1.5.4.2082, Watertown, MA, USA). Therefore, not only were strength parameters analyzed but also time-based parameters, as these provide more information about neuromuscular performance [36]. To provide a detailed overview of the overall muscle performance of the lower limb, braking, propulsion, and flight phases were also analyzed. In addition, the parameters used in a previous CMJ study for differentiation between pwMS without motor disability and HC were selected for this study [14]. Table 1 shows a description of the recorded jump parameters according to the jump phases.

### 2.4. Statistical Analysis

Force values were converted (normalized) to values relative to body weight (N/kg). The distribution of all jump parameters was visually inspected and supplemented with the Shapiro–Wilk test for the assessment of normality. In the evaluation of jump parameters, a descriptive specification of mean values and standard deviations occurred. To evaluate associations of CMJ performance, disease condition (pwMS, HC), age, sex, and BMI, and to quantify differences in jump performance between the associated subgroups (Section 2.2.1), generalized linear models (GLM) were applied. The (continuous) jump parameters served as dependent variables, and fixed main effects were quantified for the following independent variables in each multivariable model: group (pwMS, HC), age (young, middle-aged, old), sex (male, female), BMI (normal weight, overweight, obesity), and the interactions of the group variable with age, sex, and BMI (e.g., age*group, sex*group, and BMI*group). In particular, multiple (nested) GLMs were constructed a priori with a focus on simplicity; starting with a minimal adequate model, including the pre-specified independent variables of interest (basic model 1: group, age, sex, BMI), we progressively added the interaction terms (model 2: model 1 variables and the age*group interaction; model 3: model 1 variables and the sex*group interaction; model 4: model 1 variables and the BMI*group interaction). Thus, model 1 is nested within models 2–4 (and subsequent ones), and accordingly, removal of (insignificant) interaction terms results in simpler models (e.g., basic model 1). Gaussian distribution with an identity link was used for normally distributed jump parameters, and Gamma distribution with a log link function was used for right-skewed jump parameters. Pairwise comparisons (post hoc tests) were conducted, and adjustments were made using the Bonferroni correction to account for multiple group comparisons. Statistical significance was fixed at *p* < 0.05. Effect sizes were determined using eta squared and were interpreted as a measure of effect size as small (η^2^ > 0.01), medium (η^2^ > 0.06), or large (η^2^ > 0.14) [37]. The statistical analyses were performed using IBM Statistical Package for the Social Sciences (SPSS) for Windows, Version 28 (IBM Corp, Armonk, NY, USA).

## 3. Results

### 3.1. Participants

A total of 262 study participants (164 pwMS, 98 HC) were eligible for analysis. There was an age distribution of 18 to 64 years among the study participants, with a mean age of 36.29 years (SD ± 9.74) and a mean BMI of 24.59 (SD ± 4.33). In the study cohort, 64.4% were females. Most participants were middle-aged (all: 61.8%; pwMS: 64%; HC: 58.2%) and had normal weight (all: 63.4%; pwMS: 60.4%; HC: 68.4%) (see Table 2). No significant differences for sex, age, and the BMI were found between pwMS and HC. An overview of the participants’ characteristics is shown in Table 2.

### 3.2. Effect of Group, Age, Sex, and BMI on CMJ Parameters

The independent variables—age, gender, and BMI (factors)—had significant individual effects on the dependent variables (jumping parameters) (model 1; Table 3); however, each of their combined effects (i.e., the interaction of age*group, sex*group, and BMI*group) was not significant (model 2–4, Table 3). Table 3 shows the results of the associations of CMJ performance (jump parameter), group (pwMS, HC), age, sex, and BMI from the GLM analysis. The results of the contrast (post hoc) tests for group, age, sex, and BMI for model 1 (both MS and HC) are shown in the Figure 2, Figure 3, Figure 4 and Figure 5.

#### 3.2.1. Effect of Group

A significant group effect was shown for all performance parameters, force at zero velocity (FZV), peak force, negative power, and flight time, with the highest effect size, with a moderate effect for power and force, in the braking phase (FZV: η^2^ = 0.0818; and negative power: η^2^ = 0.0709) (Table 3 and Appendix A).

Overall, pwMS had significantly lower peak force in both the braking and propulsive phases compared to HC (see Figure 2). In addition, a significantly lower jump height, reactive strength index (RSI), flight time–contraction time ratio (FTCTR), and negative power, as well a shorter flight time, were observed in the pwMS (see Figure 2). The contrast (post hoc) tests for each group are shown in the Figure 2. Appendix A presents a dataset that provides a descriptive analysis of CMJ performance in association to age, sex, and BMI in pwMS and HC.

For young and middle-aged participants (18–49 years), significant group differences (pwMS; HC) were found for almost all jumping parameters (expect braking time, propulsive time, and positive power). In contrast, in the 50–65 age, group differences only exist for FTCTR. There were significant group differences in all three BMI categories for flight time and negative power. For FZV, peak force, RSI, and jump height, pwMS and HC with normal weight and overweight are significantly different. In addition, significant group differences are shown for all jump parameters (except braking time, propulsion time, and positive power) for both females and males. The group comparison between pwMS and HC in terms of age, BMI, and sex categories is shown in Table 4.

#### 3.2.2. Effect of Age

Significant age effects were found for all performance parameters, peak force, negative power, positive power, and flight time (Table 3 and Appendix A). A significant decrease in jumping performance with increasing age was observed for all participants (pwMS and HC) for peak force, positive power, flight time, RSI, and jump height. Furthermore, a significant decrease in jumping performance was shown between young and old participants as well as between middle-age and old for negative power and FTCTR. The contrast (post hoc) tests for age on the jumping parameters for all participants (both MS and HC) are shown in the Figure 3.

#### 3.2.3. Effect of Sex

For sex, significant effects were found for all performance parameters, i.e., peak force, negative power, positive power, and flight time, with the highest effect for the performance parameter in the flight phase (flight time: η^2^ = 0.2326; and jump height: η^2^ = 0.2336) (Table 3 and Appendix A). In all participants (pwMS and HC), males showed significantly higher peak force, higher positive and negative power, and longer flight time than females. Additionally, males demonstrated significantly higher jump performance in all performance parameters. The contrast (post hoc) tests for sex on the jumping parameters for all participants (both MS and HC) are shown in the Figure 4.

#### 3.2.4. Effect of BMI

Significant BMI effects were found for positive power, negative power, RSI, FZV, FTCTR, jump height, and flight time (Table 3 and Appendix A). Similar to sex, jump height (η^2^ = 0.0802) and flight time (η^2^ = 0.0828) had the largest effect size for BMI, with a moderate effect. A significant decrease in jumping performance with increasing BMI was observed for all participants (pwMS and HC) for positive power, flight time, RSI and jump height. Additionally, a significant decrease in jump performance between normal weight and obesity and between overweight and obesity were found for FZV and negative power. The contrast (post hoc) tests for BMI on the jumping parameters for all participants (both MS and HC) are shown in the Figure 5.

#### 3.2.5. Association of Age, Sex, and BMI on the Jumping Parameters in pwMS

For age in pwMS, a significant decrease in jumping performance was observed between young and old participants, as well as middle-age and old participants, for jump height, flight time, RSI, and positive power (see Appendix A). FTCTR, peak force, and negative power decreased significantly from middle-age to old age. Comparing the sex differences in jump performance in pwMS, males showed significantly better CMJ performance than females, especially for the kinetic and power parameters (see Appendix A). For BMI, the pwMS showed a significant decrease in jump performance from normal weight to overweight and from normal weight to obesity for flight time, force at zero velocity, positive power, and all performance parameters (see Appendix A). The contrast (post hoc) tests for age, sex, and BMI on the jumping parameters for pwMS are shown in the Appendix A.

## 4. Discussion

In this study, we analyzed the CMJ performance in HC and pwMS in association with age, sex, and BMI. Furthermore, reference values for the CMJ for both the pwMS and HC were provided. Our study results demonstrated that age, sex, and BMI are associated with the CMJ performance in pwMS and HC. Overall, pwMS demonstrated reduced CMJ performances compared to HC in all performance parameters, i.e., force at zero velocity, peak force, negative power, and flight time. Especially in the middle-age group (age 31–49 years), the pwMS showed significantly lower CMJ performance in all parameters compared to HC. Significant group differences (pwMS; HC) in most parameters were observed for normal weight and overweight (CMJ performance for pwMS < HC). In addition, the pwMS showed reduced CMJ performance in almost all jump parameters (with the exception of braking time, propulsion time, and positive power) for both females and males. Significant age, sex, and BMI effects were found for all performance parameters, flight time, and negative and positive power for pwMS and HC. With increasing age and BMI, CMJ performance decreased and was higher in males (both in pwMS and HC). No significant interaction effects (e.g., age*group, sex*group, and BMI*group) were found. These results suggest that the association of age, sex, and BMI on CMJ performance is almost the same for pwMS and HC and not specific to pwMS. One reason for this could be that the two groups did not differ in terms of age, sex, and BMI (Table 2).

As expected, males showed a higher CMJ performance than females. The highest significant effects were found for sex on flight time (η^2^ = 0.23), jump height (η^2^ = 0.23), and positive power (η^2^ = 0.13). In pwMS, the males jumped 40% higher than females (HC: 38%). Similar results were found in other studies [20,23,24,25,38]. Sex differences are attributed to the normal growth between males and females. Males typically have a higher percentage of fast twitch (type II) fibers, which are responsible for explosive movements and high muscle strength [39,40,41]. There are numerous other contributors to differences between the sex, including hormones, chromosomes, and gene-by-environment interactions (e.g., epigenetics) [41]. Besides sex and age, MS also seems to negatively influence the skeletal muscle fiber cross-sectional area, muscle strength, and muscle mass of the lower limbs of mildly affected MS patients, independent of MS type and disease severity [9].

In addition, older pwMS had significantly lower CMJ performance than younger pwMS. These age differences were more pronounced for performance and kinetic parameters than for temporal parameters. Similar results were found where, with advanced age, lower extremity muscle power became reduced to a comparable extent in pwMS and HC [2]. Our finding that old pwMS and HC (50+) do not differ significantly in jumping performance in most parameters (only in FZV) are consistent with Kirkland et al. [42]. They also showed no differences in jumping performance between pwMS and old HC on most measures of jumping, suggesting that the old HC also have some early deficits in neuromuscular control of jumping [42]. In addition to structural, biochemical, and cellular changes, skeletal muscle also undergoes functional changes that negatively affect its mechanical function [43]. Studies have shown that muscle strength may be a more important factor than muscle mass in determining functional limitations and mobility status with age, resulting in a decrease in jumping performance [44]. Vertical jump performance decreases with age due to loss of muscle mass and increase in body fat [45]. It is important to differentiate the normal ageing process, age-related diseases, or an increased degenerative process directly related to the progression of MS. Most often, a combination of these factors leads to the progression of neuromuscular symptoms [46]. To reduce the influence of MS disability and focus on other influencing variables such as age, only MS patients with a low EDSS (<3.0) were included in our study. The median EDSS score of 1.5 in pwMS showed that the MS cohort had only minimal signs in more than one functional system but not obvious disability [31].

The results of a large sarcopenia study showed that power-based measurements, such as leg extension strength dynamometer and the sit-to-stand test, already started to decline at age +50 years. However, power-based parameters such as handgrip strength, habitual gait speed, and lean mass remained unaltered until after the age of +70 years. The cut-off values obtained in this study differ from previous reference data, emphasizing the significance of obtaining updated and local reference materials [7].

Such cut-off values and reference values will also be useful for the CMJ in MS to identify early individual deficits in neuromuscular and muscle mechanical function in the future. Our dataset provides the first CMJ characterization in association with age, sex, and BMI in MS which can used for normative reference in MS. In future studies, a larger MS cohort is needed to generate standardized and reliable normative references that are clinically and scientifically relevant in MS.

Our results showed that a high BMI negatively influenced jumping performance in pwMS. The review by Gianfrancesco and Barcellos also found a significant association between obesity and MS [47]. Thus, obesity at the onset of MS is associated with higher disability scores and a more rapid increase in disability over an observation period of up to 6 years [48]. Obesity is a modifiable risk factor whose influence on disease onset and progression is critical for exercise therapy. Also, the significant group (HC and pwMS) differences, especially in the young participants (between 18 and 30 years), indicate the clinical relevance of the importance of early neurorehabilitation. The CMJ can help detect subtle impairments and provide individualized neurorehabilitation to positively impact functional and neurological reserve [14,49]. The therapeutic benefit of strength training to improve muscle strength and functional capacity is well recognized in pwMS [50]. High-intensity resistance or strength training combined with functional tasks is recommended in this case [51,52].

Several sport-specific studies have shown that high levels of physical activity have a positive effect on jumping performance in adults [19,22]. However, no studies have investigated physical activity in relation to jumping performance in pwMS. This would be important for future studies investigating the association of CMJ performance and physical activity in MS. Based on the knowledge that muscle strength correlates strongly with vertical jump height, it is crucial for further studies to explore this correlation with dynamometer measurements while considering the confounding variables in pwMS [53]. Other confounding variables, which may be a combination of demographic, clinical, and imaging findings, as well as biomarkers, should be investigated in future studies to obtain a complex characterization of mechanical muscle function using the CMJ in pwMS. In addition, longitudinal studies are needed to better understand the influence of the investigated variables and to determine whether the CMJ can be used as a standardized measure of disease progression in MS.

## 5. Conclusions

In conclusion, the study results showed a significant association between age, sex, and BMI on CMJ performance in pwMS and HC. Our results also suggest that the association of age, sex, and BMI on CMJ performance is almost the same for pwMS and HC. However, pwMS showed significantly lower CMJ performance than HC in middle-age participants (31–49 years), normal weight to overweight, and in both women and men. Characterizing CMJ in association with age, sex, and BMI is useful for identifying individual deficits in neuromuscular and muscle mechanical function in pwMS. These results may be useful in the development of reference jump values and digital twins in order to decisively advance the necessary implementation of individualized neurorehabilitative management of MS at an early stage [27]. In order to determine the causal influence of age, sex, and BMI on mechanical muscle function in pwMS, future studies will need longitudinal data that include other influencing factors such as physical activity and additional assessments such as the dynamometer.

## Figures and Tables

**Figure 1 biomedicines-12-00971-f001:**
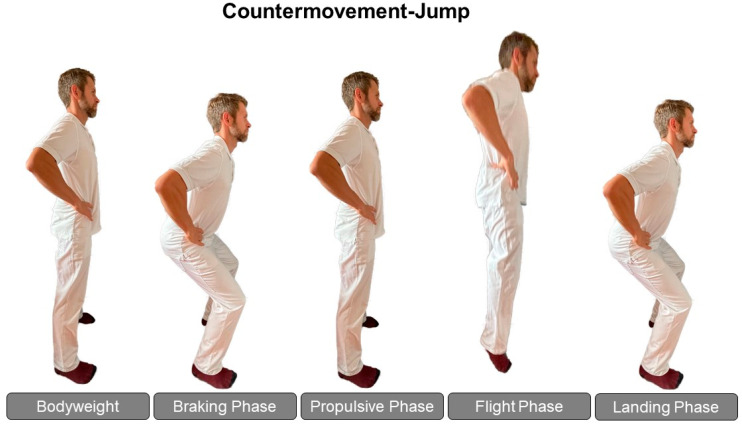
Counter movement jump technique and phases. Bodyweight: the patient stands on the force plate while their body weight is measured. During the braking phase, the patient squats down until their center of mass reaches its lowest point and their velocity is zero. Propulsive phase: the patient pushes up from the squat and the center of mass velocity becomes positive; flight phase: the time after take-off to the highest point of the center of mass; landing phase: The initial position is reached again when the patient stands still on the force plate, with both feet touching the plate.

**Figure 2 biomedicines-12-00971-f002:**
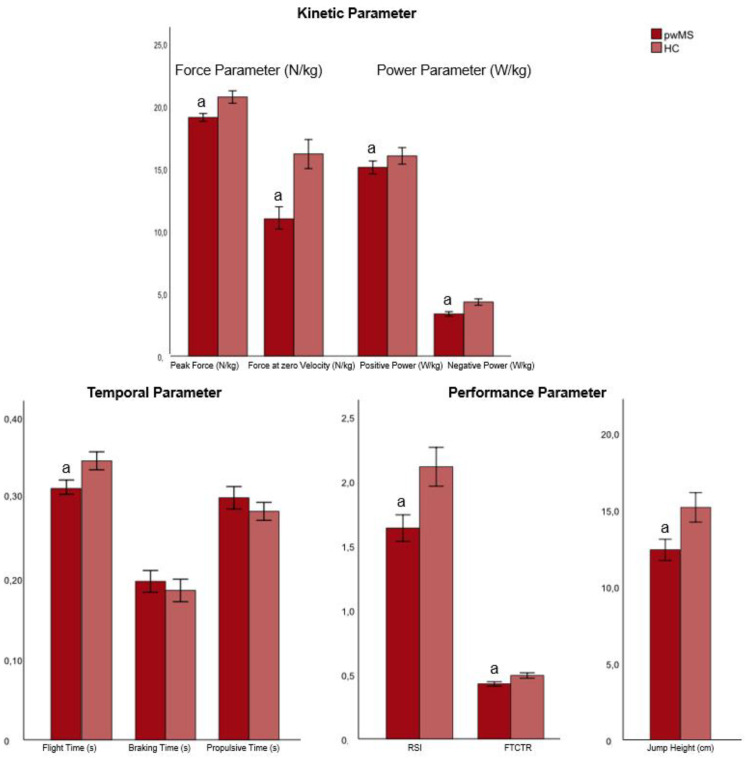
Contrast (post hoc) tests for group (pwMS and HC) on the jumping parameters (means ± SD) (model 1). Abbreviations: a = significant difference with HC (*p* < 0.05); RSI = reactive strength index; FTCTR = flight time–contraction time ratio; RSI and FTCTR have an arbitrary unit.

**Figure 3 biomedicines-12-00971-f003:**
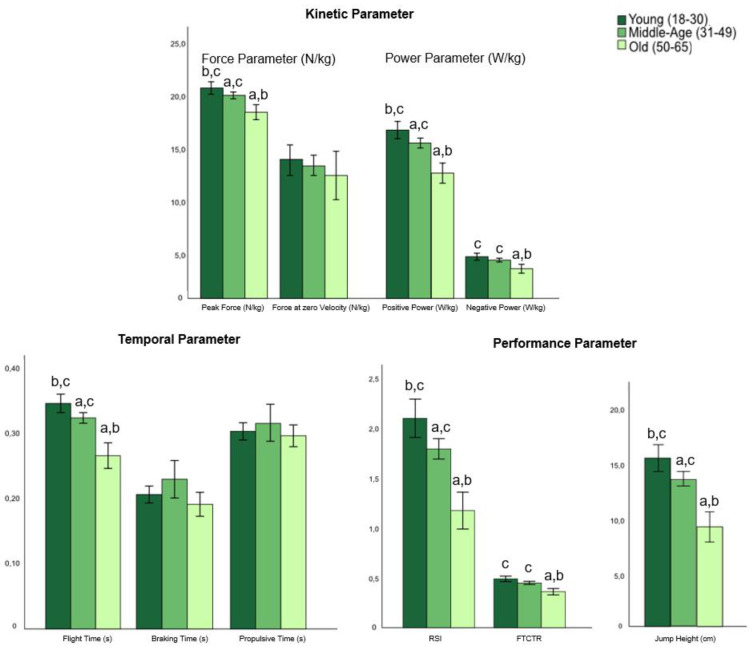
Contrast (post hoc) tests for age (young, middle-age, old) on the jumping parameters (means ± SD) for all participants (both MS and HC) (model 1). Abbreviations: a = significant difference to young (*p* < 0.05); b = significant difference to middle-age (*p* < 0.05); c = significant difference to old (*p* < 0.05); RSI = reactive strength index; FTCTR = flight time–contraction time ratio, RSI and FTCTR have an arbitrary unit.

**Figure 4 biomedicines-12-00971-f004:**
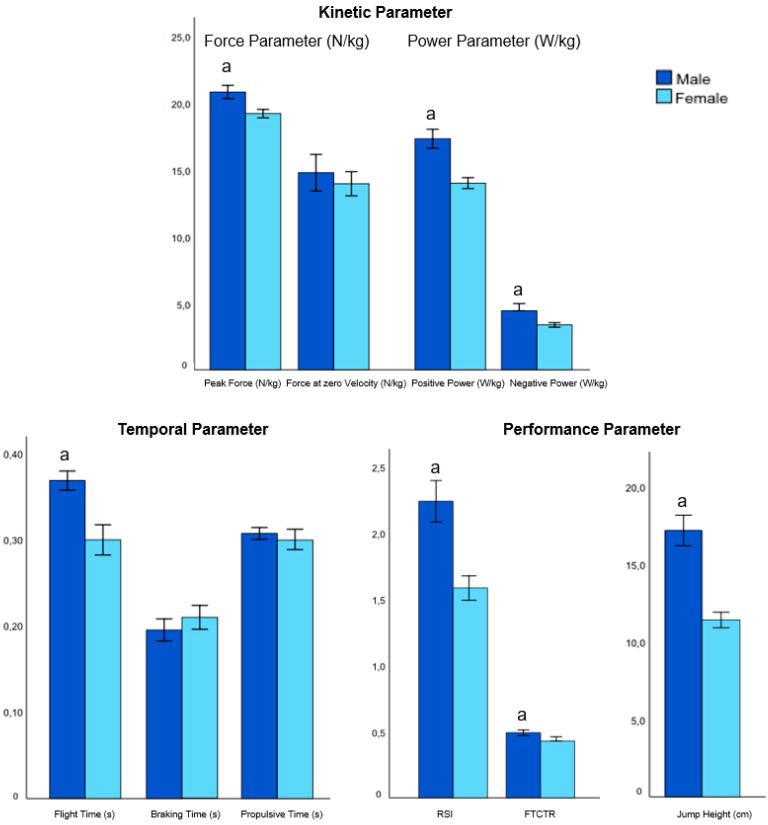
Contrast (post hoc) tests for sex (male, female) on the jumping parameters (means ± SD) for all participants (both MS and HC) (model 1). Abbreviation: a = significant difference with female (*p* < 0.05); RSI = reactive strength index; FTCTR = flight time–contraction time ratio, RSI and FTCTR have an arbitrary unit.

**Figure 5 biomedicines-12-00971-f005:**
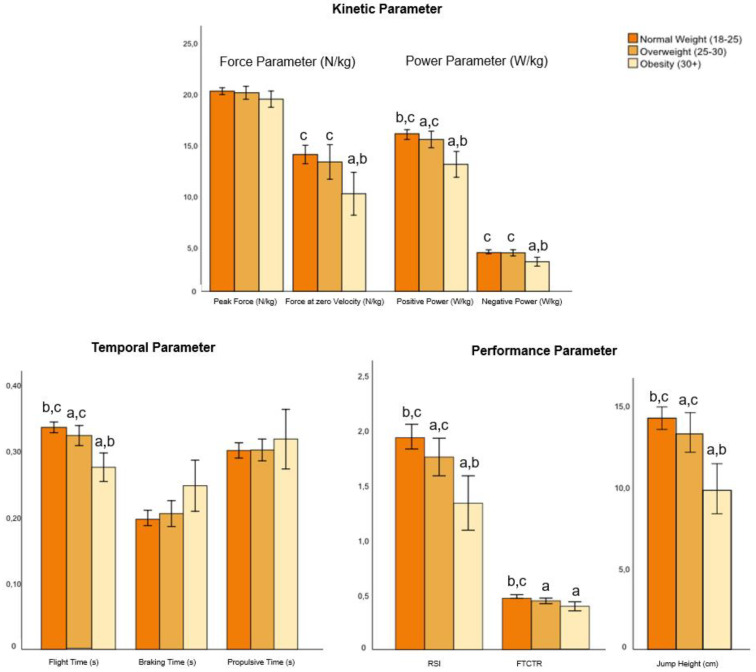
Contrast (post hoc) tests for BMI (normal weight, overweight, obesity) on the jumping parameters (means ± SD) for all participants (both MS and HC) (model 1). Abbreviations: a = significant difference to normal weight (*p* < 0.05); b = significant difference to overweight (*p* < 0.05). c = significant difference to obesity (*p* < 0.05); RSI = reactive strength index; FTCTR = flight time–contraction time ratio; RSI and FTCTR have an arbitrary unit.

**Table 1 biomedicines-12-00971-t001:** Countermovement jump parameters measured with the force plate and assignment to jump phases.

Variable Type	Parameters	Unit	CMJ—Phase
Temporal Parameters	Braking Time	s	Braking Phase
	Propulsive Time	s	Propulsive Phase
	Flight Time	s	Flight Phase
Kinetic Parameters	Force at Zero Velocity	N/kg	Braking Phase
	Peak Force	N/kg	Propulsive Phase
	Negative Power	W/kg	Braking Phase
	Positive Power	W/kg	Propulsive Phase
Performance Parameters	Jump Height	cm	Flight Phase
	Flight Time–Contraction Time Ratio	-	Contraction/Flight Phase
	Reactive Strength Index	-	Contraction/Flight Phase

Abbreviations: CMJ = countermovement jump; contraction phase = total of braking and propulsive phase.

**Table 2 biomedicines-12-00971-t002:** General characteristics of participants. Data are presented as mean (±standard deviation) unless specified otherwise.

		HC (N = 98)	pwMS (N = 164)	*p*-Value
Sex (female) N (%)		59 (60.2)	110 (67.1)	0.161 ^a^
Age (years)		36.60 ± 10.56	36.10 ± 9.25	0.699 ^b^
Age Categories	Young (18–30) N (%)	30 (30.6)	44 (26.3)	
	Average Age	25.40 ± 3.26	25.95 ± 3.94	0.527 ^b^
	Middle-age (31–49) N (%)	57 (58.2)	105 (64)	
	Average Age	38.56 ± 5.59	37.72 ± 5.50	0.359 ^b^
	Old (50–65) N (%)	11 (11.2)	15 (9.1)	
	Average Age	57.00 ± 4.36	54.53 ± 4.17	0.157 ^b^
BMI		24.37 ± 3.96	24.71 ± 4.55	0.549 ^b^
BMI Categories	Normal weight (18–25) N (%)	69 (68.4)	99 (60.4)	
	Average BMI	22.35 ± 1.76	21.86 ± 1.86	0.086 ^b^
	Overweight (25–30) N(%)	43 (26.2)	22 (13.4)	
	Average BMI	26.70 ± 1.33	26.91 ± 1.49	0.587 ^b^
	Obesity (30+) N (%)	9 (9.2)	22 (13.4)	
	Average BMI	33.87 ± 3.14	33.25 ± 3.92	0.680 ^b^
Disease Duration (years)		n.a	6.87 ± 5.26	-
MS Subtype				
	RRMS (%)	n.a	100%	-
EDSS (median, IQR)		n.a	1.5 (1.5–2.0)	-
	Pyramidal FSS	-	0 (0–1)	-
	Cerebellar FSS	-	0 (0–0)	-
	Ambulation	-	0 (0–0)	-

Abbreviations: ^a^ = calculated with chi^2^, ^b^ = calculated with *t*-test, pwMS = people with multiple sclerosis; HC = healthy controls; MS = multiple sclerosis; RRMS = relapsing remitting multiple sclerosis; EDSS = expanded disability status scale; BMI = body mass index; IQR = interquartile range; FSS = functional system score, n.a. = not applicable.

**Table 3 biomedicines-12-00971-t003:** Effect of group, age, BMI, and sex on the CMJ performance.

	Model 1				Model 2	Model 3	Model 4
Parameters (Dependent Variable)	Group (MS, HC)	Age Categories (Young, Middle-Age, Old)	BMI Categories (Normal Weight, Overweight, Obesity)	Sex Categories (Male, Female)	Interaction Group*Age Categories	Interaction Group*BMI Categories	Interaction Group*Sex Categories
Braking Time ^1,A^	NS	NS	NS	NS	NS	NS	NS
	-	-	-	-	-	-	-
Propulsive Time ^1,A^	NS	NS	NS	NS	NS	NS	NS
	-	-	-	-	-	-	-
Flight Time ^1,B^	*p* < 0.001 *	*p* < 0.001 *	*p* < 0.001 *	*p* < 0.001 *	NS	NS	NS
	η^2^ = 0.0478	η^2^ = 0.0505	η^2^ = 0.0828	η^2^ = 0.2326	-	-	-
FZV ^2,B^	*p* < 0.001 *	NS	*p* = 0.002 *	NS	NS	NS	NS
	η^2^ = 0.0818	-	η^2^ = 0.0275	-	-	-	-
Peak Force ^2,A^	*p* < 0.001 *	*p* = 0.004 *	NS	*p* < 0.001 *	NS	NS	NS
	η^2^ = 0.0540	η^2^ = 0.0233	-	η^2^ = 0.0053	-	-	-
Negative Power ^2,A^	*p* < 0.001 *	*p* = 0.045 *	*p* = 0.004 *	*p* < 0.001 *	NS	NS	NS
	η^2^ = 0.0709	η^2^ = 0.0133	η^2^ = 0.0235	η^2^ = 0.0526	-	-	-
Positive Power ^2,A^	NS	*p* < 0.001 *	*p* < 0.001 *	*p* < 0.001 *	NS	NS	NS
	-	η^2^ = 0.0381	η^2^ = 0.0556	η^2^ = 0.1313	-	-	-
Jump Height ^3,B^	*p* < 0.001 *	*p* < 0.001 *	*p* < 0.001 *	*p* < 0.001 *	NS	NS	NS
	η^2^ = 0.0481	η^2^ = 0.049	η^2^ = 0.0802	η^2^ = 0.2336	-	-	-
FTCTR ^3,B^	*p* = 0.005 *	*p* = 0.003 *	*p* = 0.003 *	*p* = 0.006 *	NS	NS	NS
	η^2^ = 0.0327	η^2^ = 0.0248	η^2^ = 0.0248	η^2^ = 0.0318	-	-	-
RSI ^3,A^	*p* < 0.001 *	*p* < 0.001 *	*p* < 0.001 *	*p* < 0.001 *	NS	NS	NS
	η^2^ = 0.0520	η^2^ = 0.0403	η^2^ = 0.0506	η^2^ = 0.0914	-	-	-

Results based on the GLM analysis. ^1^ = temporal parameter; ^2^ = kinetic parameter; ^3^ = performance parameter; ^A^ = Gamma distribution with log link function; ^B^ = Gaussian distribution with identity link; Model 1 = group, age, sex, and BMI; Model 2 = group, age, sex, BMI, and group*age interaction; Model 3 = group, age, sex, BMI, and group*BMI interaction; Model 4 = group, age, sex, BMI, and group*sex interaction; * = significant effect (*p* < 0.005); η^2^ > 0.01 = small effect; η^2^ > 0.06 = medium effect; η^2^ > 0.14 = large effect. Abbreviations: NS = not significant; FZV = force at zero velocity; RSI = reactive strength index; FTCTR = flight time–contraction time ratio.

**Table 4 biomedicines-12-00971-t004:** Group comparison between pwMS and HC in terms of age, BMI, and sex.

	Age Categories			BMI Categories			Sex	
	Young 18–30 (pwMS = 44, HC = 30)	Middle-Age 31–49 (pwMS = 105, HC = 57)	Old 50–65 (pwMS = 15, HC = 11)	Normal Weight 18–25 (pwMS = 99, HC = 69)	Overweight 25–30 (pwMS = 43, HC = 22)	Obesity 30+ (pwMS = 22, HC = 9)	Male (pwMS = 54, HC = 39)	Female (pwMS = 110, HC = 59)
Parameters	*p*	*p*	*p*	*p*	*p*	*p*	*p*	*p*
BT ^1^	0.097	0.002 *	0.853	0.146	0.141	0.184	0.069	0.133
PT ^1^	0.348	0.005 *	0.655	0.010 *	0.233	0.846	0.058	0.309
FT ^1^	<0.001 *	<0.001 *	0.174	0.003 *	<0.001 *	0.020 *	<0.001 *	<0.001 *
FZV ^2^	<0.001 *	<0.001 *	0.039 *	<0.001 *	<0.001 *	0.057	0.003 *	0.001 *
PF ^2^	<0.001 *	<0.001 *	0.506	0.005 *	<0.001 *	0.177	<0.001 *	0.030 *
NP ^2^	<0.001 *	<0.001 *	0.064	0.001 *	0.001 *	0.020 *	<0.001 *	0.001 *
PP ^2^	0.051	<0.001 *	0.316	0.078	0.047 *	0.114	0.007 *	0.187
JH ^3^	<0.001 *	<0.001 *	0.187	0.002 *	<0.001 *	0.029 *	<0.001 *	0.002 *
FTCTR ^3^	0.002 *	<0.001 *	0.480	0.012 *	0.003 *	0.233	<0.001 *	0.048 *
RSI ^3^	<0.001 *	<0.001 *	0.329	0.007 *	<0.001 *	0.087	<0.001 *	0.011 *

Abbreviations: * = significant (*p* < 0.05); ^1^ = temporal parameters; ^2^ = kinetic parameters; ^3^ = performance parameters; pwMS: people with multiple sclerosis. HC: healthy controls; BMI = body mass index; BT = braking Time; FZV = force at zero velocity; NP = negative power; PF = peak force; PP = positive power; JH = jump height; FT = flight time; FTCTR = flight time–contraction time ratio; RSI = reactive strength index; PT = propulsive time.

## Data Availability

All data produced in the present study are available upon reasonable request to the authors.

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
