# Peer review of "The Association of Age, Sex, and BMI on Lower Limb Neuromuscular and Muscle Mechanical Function in People with Multiple Sclerosis"

_biomedicines, 2024, doi:10.3390/biomedicines12050971_

Round 1

Reviewer 1 Report

Comments and Suggestions for Authors

The Association of Age, Sex and BMI on Lower Limb Neuro-2 muscular and Muscle Mechanical Function in People with Multiple Sclerosis

By Glessner et al.

The paper concerns assessing effects of MS type, age, sex and BMI to various parameters measured when performing a countermovement jump.

The results of the paper are interesting and the experiment appears well carried out. However, the analytic treatment raises questions, and indeed the post analysis presentation could be improved quite a bit. In fact it must. Specific comments follow below.

The use of acronyms: this will need to be reconsidered. In the manuscript there does not appear to be a consistent use. For example, EDSS is first mentioned in line 43, where the acronym is defined. However, the concept is first spelled out again in the heading to section 2.2.2, and subsequently redefined below in line 114. The definition should appear only once. If you have a principle of spelling the concepts out in the headings in section 2.2 then all fine, but in 2.2.1 you do not do that with BMI, and in 2.2.3 the heading "Jumping performance" does not conform with the acronym (CMJ). And here, the acronym is not spelled out in the descriptive part of the section.

Please, be consistent.

Line 40: "... useful outcome..." ? useful for what? predicting the stage of disease? or? please specify.

Line 120: You mention here that out of three jumps, the mean value was used for all individual parameters were used. I wonder why you did not choose to use the max? After all, the CMJ experiment is about maximizing your efforts, as you write in lines 125-26, and the average might show something else. The max was for example chosen by Philpot et al 2021, doi 10.1177/1754337120971436 , that also studies CMJ.

Lines 135, 151: 'COM' is not defined. Do you mean CMJ?

Line 136: The 'initial position' is not defined. Is it the one in the braking phase?

Lines 175-76: What are 'jump parameters'? Do you mean all the parameters in Table 1? They are labeled Temporal, Kinetic and Performance parameters. Please explain. Also, which of these were analyzed with the normal model, and which with the Gamma model? These are radically different approaches, and it would be good to know which jump parameters that were analyzed with what. If you mean the parameters from Table 1 there's only 10 of them, so it would be easy to list them up. 

Lines 214+: Table 3. I must say that I do not like this table, and perhaps what was used to create it. As you note in line 211, none of the interactions are significant for any of the measured parameters. Yet they still appear in the table with effect measure and all. This raises the question: In which model is the p-value for to example BMI calculated? It does usually not make sense to test a main effect in a model where it also appears as part of an interaction, as the result of removing the main effect from the model does not remove the variable form the model. I have not noted anything from your section on statistical analyses that you are sequentially removing insignificant terms. Does this mean that the p-values are relative to a model where the interaction terms are in? If that is so, I will question the validity of your results. If it is not so, it should be specified in the analysis section, and p-values and effect measures in Table 3 should be replaced with a NS and a '-' for insignificant terms. 

Lines 241-47: Here the authors claim that '... tests from the GLM model of age categories in pwMS revealed, that jumping performance decreased significantly with age'. First, where is the 'significantly' phrase documented? The authors start talking about the temporal parameters right after this general statement, but of these, it is only flight time that is significant (that is, if we subsume that you values in Table 3 are valid). Further, you have chosen to describe the impact of the categories through eta^2, and with that you also leave out the sign of the effect, where the reader is left to work out where you can find it. It appears that these are what is in your appendix, to which you should refer. However, I disagree that the authors have shown a decrease in flight time with age. In fact your Figure 2 also shows that there does not appear to be a difference between young and middle-aged individuals. It is only when the authors compare to 'Old' that the significance appears to manifest, and that is something else. If we follow from the statement, then the flight time should go down if you get older. But if your age increases from 18 years to 48 years, there is no evidence in the data (still subsuming that the estimates are valid) that the flight time decreases. The authors lists the significances right after this statement, but fail to mention that there does not appear to be a significance between young and middle-aged. Last, the authors write that the significant decrease with age is for pwMS. No, it is not. It is not, because you do not have any significant interaction term between group and Age for the flight time model. And that means that the effect of age is the same for both HC and pwMS; ie. not an effect that is particular for pwMS. It is just a general decline with (progressed) age that you have found.

Line 248: Figure 2. Here, values for the three age groups are listed together with a version of the compact letter display. However, the use of compact letter display (a, b and c) is limited to show significances, which invites misinterpretations like the above. I can recommend using compact letter display in the standard way, which is specified on t. ex. Wikipedia. I will also question the graphics. What is the level of the bars? The authors write that it is 'for pwMS', and talk about that the association between age and jumping parameters are based on the GLM model. But there is only one bar for each parameter and, say, young individuals, despite these individuals having varying sex and varying BMI. What is it then? Is it a weighted average? Or is it empirical data? If it is a weighted average, then it should be written. If it is empirical data, then what is the impact of the model in the figure? The confidence intervals? The compact letter display? If the authors just want to visualize the groupings based on the model, they could reduce the model to not contain interaction terms (if that is possible), and then just choose a reference category, say pwMS, males and normal weight, because the classification is irrespective of the class that it is displayed for when there are no interactions. Same goes for Figures 3 and 5.

Lines 281 and 283. I believe that the authors overstate the evidence of the study here when they write that 'age, sex and BMI are associated with the CMJ performance in pwMS and HC'. While it is formally correct, the authors do not mention that there is no interaction with group for any of the three measures, and that means that none of the effects are specific to pwMS. From their formulation that is the immediate thought that you get, and there is absolutely no evidence presented for that. The sentence in lines 291-292, 304-305 and others have the same impact. The effects of age, gender and BMI appears to be there, but they do not appear to be associated with pwMS in any way.

All in all, the statistical treatment of the material is insufficient and (sorry but that is how I read it) misleading. The analysis should be redone, more details provided, and the form of the presentation of data and results should be reconsidered. 

Author Response

Dear reviewer,

We would like to thank you for your valuable comments on our manuscript. We have considered these comments carefully and have provided a response to each in full below. We believe that the comments have strengthened the paper.

Reviewer 2 Report

Comments and Suggestions for Authors

ABSTRACT: The abstract could benefit from a clearer articulation of the specific research question or objective. Although it mentions investigating the association of age, sex, and BMI with muscle mechanical function in pwMS, a more explicit statement of the research aim would enhance clarity for readers.

While the abstract mentions significant effects of age, sex, and BMI on CMJ performance, it could be strengthened by providing more specific statistical insights or effect sizes. This would help readers better understand the magnitude and importance of these associations.

The abstract lacks a clear statement about the potential implications or applications of the findings. Adding a brief sentence about how understanding these associations could enhance clinical assessment or inform interventions for people with MS would increase the relevance and importance of the study.

INTRODUCTION: The introduction presents a wealth of information without a clear structure to guide the reader through the presented concepts. It would be beneficial to rearrange the text to more orderly and sequentially present information about the symptoms and pathological mechanisms of multiple sclerosis, followed by the importance of evaluating muscle function in patients, and finally, introducing the relevance of CMJ as a novel assessment tool in this context. Although several factors influencing muscle function and CMJ performance are mentioned, the introduction does not clearly establish how these factors relate to the specific study objective. It would be helpful to more explicitly connect the discussion on the influence of age, sex, and BMI on muscle function with the need to characterize CMJ performance in people with multiple sclerosis and establish reference values. This would help readers better understand the relevance and motivation of the study.

MATERIALS AND METHODS: The inclusion and exclusion criteria should be more specific and detailed to ensure proper participant selection. For example, it is mentioned that participants must have the ability to walk unaided and rest for at least 500 m, but it would be useful to specify if there are any additional restrictions in terms of walking speed or specific mobility conditions that could affect the ability to perform the jump. The description of the data collection process on CMJ performance is quite detailed, which is positive. However, it would be beneficial to provide additional information about the exact setup of the equipment used, including calibration parameters and any methods used to minimize measurement errors. It is mentioned that certain CMJ performance parameters were selected based on a previous study, but it would be helpful to provide a more explicit justification of why these specific parameters were chosen and how they relate to the objectives and hypothesis of the current study. This would help readers better understand the relevance of the selected parameters and their application in the context of the study.

RESULTS: Although detailed results of comparisons between groups of people with multiple sclerosis (pwMS) and healthy controls (HC) are presented, the interpretation of these results could be improved by providing clearer contextualization. It would be helpful to discuss how these findings relate to existing literature on muscle function in people with multiple sclerosis and how they contribute to our overall understanding of the disease.

DISCUSSION: Although the results obtained in relation to age, sex, and BMI in people with multiple sclerosis (pwMS) and healthy controls (HC) are described in detail, the discussion could be improved by providing broader contextualization. It would be helpful to discuss how these results relate to previous studies and how they contribute to our current understanding of muscle function in pwMS.

Although some possible explanations for the observed differences in CMJ performance among different age, sex, and BMI groups are mentioned, the discussion could benefit from a more detailed exploration of potential underlying mechanisms. For example, a more in-depth discussion could be held on the effects of decreased muscle mass, increased body fat, and other age and disease-related changes on muscle function. It would be helpful to discuss how confounding variables were handled and controlled in the study. For example, addressing how the possible effects of medication, physical activity, and other concurrent medical conditions that could have influenced the results were controlled. The discussion could be enriched by providing more specific and detailed suggestions for future research in this field. For example, additional studies could be proposed to explore the relationship between CMJ performance and physical activity in pwMS, as well as longitudinal research to assess disease progression over time.

CONCLUSION: The conclusion could benefit from a more extensive expansion on the clinical and practical importance of the results obtained. For example, discussing how characterizing CMJ performance in pwMS in relation to age, sex, and BMI can influence disease management and treatment strategies, as well as inform the design of specific rehabilitation interventions. Although the need for longitudinal data in future studies to determine the causal influence of age, sex, and BMI is mentioned, the conclusion could more specifically clarify how this data could contribute to future research in the field of multiple sclerosis. Providing some more concrete suggestions on study designs and variables that should be considered in future longitudinal research would be useful.

In general, ensure that percentages are connected to numbers. ensure consistency throughout.

Author Response

Dear reviewer,

We would like to thank you for your positve and valuable comments on our manuscript. We have considered these comments carefully and have provided a response to each in full below. We believe that the comments have strengthened the paper.

Reviewer 3 Report

Comments and Suggestions for Authors

The article is original and very relevant for the field. The authors studied the association of age, sex and BMI on muscle mechanical function in people with multiple sclerosis (pwMS).

The results showed that, PwMS showed significantly lower CMJ performance compared to health control in middle-aged, with normal weight to overweight and in both women and men. Significant age, sex and BMI effects were found for all performance parameter, flight time, negative and positive power.

The methology of the study is modern and very complex.

The conclusions are consistent with the evidence and arguments presented. This study showed that age, sex and BMI are associated with muscle mechanical function in pwMS and HC. These results may be useful in developing reference values for CMJ which is an important step towards integrating CMJ into the diagnostic assessment of people with early MS.

The references are appropriate, including some relevant authors experience in the field.

I recommend some minor corrections.

-line 60 BMI- give the detailed name at first usage

-line 85 Characterisation, correct is Characterization

-line 88 characterise, correct is characterize

Author Response

Dear reviewer,

We would like to thank you for your very ositive and valuable comments on our manuscript. We have considered these comments carefully and have provided a response to each in full below. We believe that the comments have strengthened the paper.

Round 2

Reviewer 1 Report

Comments and Suggestions for Authors

The manuscript has improved, but the statistics and the messages derived from it still give me serious causes for concern. These are specified below. The authors should argue for their approach and/or make necessary changes.

Line 40: "... useful outcome..." ? It appears that my comment was not understood by the authors. You should describe what the outcome should be used for since it is 'useful'. This is a minor issue and the authors should react as they see fit.

Table 3: I do not find the solution presented by the authors super nice. If one of the interaction terms proved significant, then what? It would render the model 1 insufficient, and consequently the two other tests for association would be illegitimate, based on incorrect models. What if two terms were in reality significant? Then all the tests would be illegitimate, based on incorrect models. Last, the factors may blur each others effects, you may overlook a significance when insiting on testing them  separately. What you in my opinion should have done would be to include all three interactions in one model, and then stepwise reducing this model down to your model one, every time removing the interaction with the highest p-value above 0.05 and wite NS for non-significance. Afterwards, assuming that you arrive at your model 1, give all the p-values there, so there is no doubt about which model the p-value refers to, or continue to reduce the model, replacing each p-value with a NS, again not leaving doubt about which model the p-value refers to. You can also choose to take the criticism from your community, but while your approach is now better than previously, I still find your argumentation for significances and non-significances vague, but easy to improve.

How ever, this leads me to my criticism of your graphics. While you have clarified what the contents of the bars and whiskers are, I still maintain my criticism that you in t.ex. your model 2, which is the basis for Figure 2, will have different sex and BMI, and because of your chosen approach you have not checked their significance IN YOUR MODEL 2. You cannot just assume that these effects aren't there if you haven't checked it. Further, you still claim that Figure 2 is 'for pwMS', which again indicates that this is something that is particular for pwMS. But the interaction term in model 2 is insignificant, and therefore there is no evidence that the difference between HC and pwMS is anything but stochastic fluctuations. When you then model the difference anyway and put this supposedly stochastic fluctuation up for use in the graphics, you indicate that the modeling supports a difference. It IS misleading to put it up like that. There is no evidence in your modeling as of now of any relation to pwMS, and when you use modeling results, you cannot use an indication of such a relation. You might want to consult a professional statistician on these matters. My suggestion will be to remove the 'for pwMS' term in Figure 2, and base yourself on t.ex. your model 1.

My expectation is that there will only be cosmetic differences to figures and tables if you make the above changes, as there really appears to be no evidence for support of relations to pwMS, but the message that the material will send will be much more in line with the information in the data.

On eta^2: The authors write that eta^2 has no sign. Indeed, that is exactly so, and it means that the sign of the change does not appear in Table 3. When you choose to use eta^2 to summarize the effects, and you then refer to that a value, say, decreases with age, you will have to reference to other documentation than Tabel 3. Both foir sign and actual magnitude. Using eta^2 is using a two-edged sword.

On the conclusion: I still find what you write misleading. It is misleading, because you say that there are effects of the variables for both pwMS and HC. But you do not write that these effects are the same in modelling terms, and when you specify the two groups you invite people to think that the effects may be different, which they absolutely aren't. Why is it so difficult to write that 'we detected no statistically significant difference in effects between pwMS and HC' here as well, when you have already written it in lines 328-331? That is apparently what the data tells you, and there is also information in that that could be discussed? I see no reference to the literature in lines 328-331, what does the literature say?

Author Response

Dear Reviewer,

We would like to thank you for your valuable comments on our manuscript. Your comments have been incredibly helpful in improving our paper. In the following comments, we will provide a more detailed explanation of our approach and highlight the additional changes we have made.

All the best

Round 3

Reviewer 1 Report

Comments and Suggestions for Authors

On Comment 2:

The authors have adressed my concerns.

On comment 3:

I find the supplementary tables with effects for the models 2-4 unnecessary. Why are they there when the right model is model 1? Also, the unclusion of tests for 'main effects' is not a good idea in my opinion. It is not clear what this test investigates, but in most software solutions, the so-called 'main effect' is the effect for the reference categories of any variables that it interacts with. Thus, if this is so in the author's analysis, the test for effect of 'group'  in Table S2 is the effect of 'group' for young people, in Table S3 it is the effect of 'group' for normal-weight, and in table S4 it is the effect of 'group' for males. Still assuming my interpretation is correct these test first of all do not carry any use, and second the doe not resemble an overall test for 'group'. I suggest that you remove the tables again, they do not add anything usable to the manuscript.

As written, You have not performed a full investigation and you might misinterpret results. If your design was pre-specified prior to analysis, you should write it under the statistical analysis section. I do not like what you have added. If as you say your models were pre-specified, you would never enter into the process you describe with multiple significant interaction terms, and indeed i can't see anywhere in the manuscript that you actually investigated it (ie. added interaction terms to models 2 and 4). It is also a bad procedure, as youo might as well use the baackwards selection process if you needed to venture that far. You will need to stick to what you did and remove this sentence; you obviously didn't plan for it, and your main argumentation for sticking to models 2-4 is that it was pre-planned. Next time you do this, my suggestion will be to to backwards selection from a saturated model.

On Comment 4:

I will repeat my advise that the authors should consult a professional statistician, which they obviosly have not. The response from the authors is a long explanation, where they essentially write that 'we do this (model 2-4 as basis for the figures) because we want to'. The authors now have noted up front that there is no interaction between group and any of the variables, so at least the observant reader will know htat there is no reason to separate out the pwMS, but following that it is rather inconsistent that they insist on doing exactly that for the figures. As also written earlier, the authors can choose to take the heat from the community now that it is at least clarified to the obervant reader that the figures are based on insignificant interactions, but in my opiniion it would be better science to replace the figures with those now in the appendix. They show the real effect in the data, and could easily be interpreted in terms of what happens when you get older and or fatter. This is said to improve the manuscript, and streamline the message that it sends.

On Comment 6:
I believe that the authors have misunderstood this. The tables they write here and elsewhere that they add does not contain the direction of the effect. However, this is so in Table S1 and the authors should refer to that table instead of tables with p-values.

On Comment 7:

The authors have adressed my concerns.

Author Response

Dear Reviewer,  
We would like to thank you for your valuable comments on our manuscript. We have considered these comments carefully and have provided a response to each in full below.  
In response to your comments, we have implemented a significant revision to the manuscript. We hope that this revised version meets your satisfaction. We are grateful for your constructive feedback and believe that this revised paper has greatly benefited from your input. 

Alle the best, 

Anne Geßner
